# Efficacy of NSCLC Rechallenge with Immune Checkpoint Inhibitors following Disease Progression or Relapse

**DOI:** 10.3390/cancers16061196

**Published:** 2024-03-18

**Authors:** Maria Effrosyni Livanou, Vasiliki Nikolaidou, Vasileios Skouras, Oraianthi Fiste, Elias Kotteas

**Affiliations:** 1Third Department of Internal Medicine, Sotiria General Hospital for Chest Diseases, National and Kapodistrian University of Athens, 11527 Athens, Greece; vknikolaidou@gmail.com (V.N.); ofiste@med.uoa.gr (O.F.); ilkotteas@med.uoa.gr (E.K.); 2Department of Pulmonary Medicine, 401 General Army Hospital, 11522 Athens, Greece

**Keywords:** immunotherapy, immune checkpoint inhibitors, rechallenge therapy, resistance, NSCLC

## Abstract

**Simple Summary:**

The majority of patients with advanced non-small-cell lung cancer (NSCLC) who initially respond to treatment with immune checkpoint inhibitors (ICIs), will ultimately develop resistance within four years. ICI rechallenge is considered in real-world practice, but its effectiveness following disease progression is not well-established. The aim of this review was to evaluate the clinical efficacy of rechallenge ICI therapy following disease progression, based on the critical assessment of the published data. The evidence shows limited efficacy of rechallenge immunotherapy in unselected patient populations who progressed during initial immunotherapy, yet promising efficacy in those who relapsed after treatment completion.

**Abstract:**

Immune checkpoint inhibitors (ICIs) are at the forefront of advanced non-small-cell lung cancer (NSCLC) treatment. Still, only 27–46% of patients respond to initial therapy with ICIs, and of those, up to 65% develop resistance within four years. After disease progression (PD), treatment options are limited, with 10% Objective Response Rate (ORR) to second or third-line chemotherapy. In this context, ICI rechallenge is an appealing option for NSCLC. Most data on the efficacy of ICI rechallenge are based on retrospective real-world studies of small, heavily pretreated, and heterogeneous patient groups. Despite these limitations, these studies suggest that ICI monotherapy rechallenge in unselected NSCLC patient populations who discontinued initial ICI due to PD is generally ineffective, with a median Progression-Free Survival (PFS) of 1.6–3.1 months and a Disease Control Rate (DCR) of 21.4–41.6%. However, there is a subpopulation that benefits from this strategy, and further characterization of this subgroup is essential. Furthermore, immunotherapy rechallenge in patients who discontinued initial immunotherapy following treatment protocol completion and progressed after an immunotherapy-free interval showed promising efficacy, with a DCR of 75–81%, according to post hoc analyses of several clinical trials. Future research on ICI rechallenge for NSCLC should focus on better patient stratification to reflect the underlying biology of immunotherapy resistance more accurately. In this review, we summarize evidence regarding rechallenge immunotherapy efficacy following NSCLC disease progression or relapse, as well as ongoing trials on immunotherapy rechallenge.

## 1. Introduction

From 2015, with the FDA approval of the first immune checkpoint inhibitor (ICI) for non-small-cell lung cancer (NSCLC) treatment [1,2] to today, immunotherapy (IO) has become the standard of care for patients with metastatic disease [3] and—following recent phase-III clinical trial results—it is expected to dominate the treatment field of locally advanced disease as well [4,5]. Despite the undeniable benefit of the introduction of immunotherapy in NSCLC treatment, only 27–46% [6,7,8] of patients with advanced disease respond to therapy with ICIs. Furthermore, even among responders, resistance to immunotherapy may eventually develop [9]. Based on a pooled analysis of four clinical trials, including patients treated with nivolumab, 65% of those who respond to treatment initially will progress within four years [10]. Beyond the point of progression, therapeutic options are limited, and the optimal management strategy is not clear. As we have gained more insight into the dynamic adaptation of cancer cells and the tumor microenvironment, the idea of retreatment with ICIs following progression after a treatment-free period is considered a reasonable strategy [11]. In this article, we perform a comprehensive review of the published literature regarding ICI retreatment of NSCLC patients following progression to previous-line immunotherapy. In the first section, we review the results of retrospective real-world studies. In the second section, we discuss the results of rechallenge immunotherapy in subsets of patients from post hoc analyses of phase-III trials, as well as a phase-II clinical trial of rechallenge immunotherapy in NSCLC. In the third section, we briefly present ongoing trials of ICI rechallenge. Finally, we briefly discuss the biological rationale of rechallenge immunotherapy and the differences in immune response between immunotherapy-naive and immunotherapy-pretreated patients and propose areas for future research.

### 1.1. Primary versus Acquired Resistance

When the prospect of an immunotherapy rechallenge is considered, it is pivotal to understand the underlying mechanism of resistance to the first course of immunotherapy. Primary resistance (also called intrinsic or innate) refers to patients who do not respond to immunotherapy; instead, their disease progresses quickly. Acquired resistance (also named secondary, adaptive, or evasive), on the other hand, refers to patients who initially responded to immunotherapy but eventually have disease progression [9,11]. Making this distinction when assessing the efficacy of immunotherapy rechallenge is important, as they indicate different predispositions to immunotherapy response and distinct mechanisms of resistance. A challenge in interpreting studies assessing the efficacy of rechallenge therapy is the lack of consistency in the clinical definition of resistance to immunotherapy. In 2020, the Society for Immunotherapy of Cancer (SITC) Immunotherapy Resistance Taskforce published its recommendations for defining resistance to PD-1 pathway blockade across solid tumors. They proposed three distinct categories of resistance based on the best overall response (BOR) and the duration of response to therapy. They defined primary resistance as progressive disease (PD) as the initial response to therapy or complete response (CR), partial response (PR), or stable disease (SD) for <6 months. Secondary resistance was defined as CR, PR, or SD for >6 months or progression within 12 weeks after the last dose of therapy for patients who completed a course of ICI regimen. Finally, patients who either completed an ICI course or stopped due to toxicity and subsequently progressed >12 weeks after the last dose were classified separately, and the expert panel proposed ICI retreatment in these patients. Several points regarding these recommendations should be noted. First of all, they are not designed for clinical decision-making but rather for the stratification of patients in clinical trials to increase the likelihood of positive results. Second, the recommendations are designed to be used across solid tumors; thus, they may not reflect the differential biology of each tumor. Third, although the expert panel has set a time limit of 6 months, they agreed that there was no strong scientific evidence to prove its validity. Finally, patients with SD are pooled with patients who have PR and CR, although research suggests differential tumor and tumor microenvironment biology in these cases [12].

More recently, Schoenfeld et al. proposed modified criteria for clinically acquired resistance, specifically in patients with NSCLC [13]. Their criteria for acquired resistance to ICI therapy were as follows: 1. Prior treatment with Immunotherapy (IO); 2. Objective response to PD(L)-1 blockade (stable disease (SD) is excluded); and 3. Progression occurring within 6 months of last PD(L)-1 blockade therapy. A potential limitation of this definition is the exclusion of all patients with the best overall response (BOR) of SD, classifying this entire population as having primary resistance. Furthermore, it does not entail a minimum duration of response to ICI therapy, as it recognizes that the 6 months proposed by SITC has not been validated in the setting of NSCLC. Yet, whether a time limit should be included in the definition in order to optimally select patients needs to be more thoroughly investigated. Like the SITC criteria, they also set a limit after the last immunotherapy dose; however, while the SITC puts this limit at 12 weeks (3 months)—based on the half-life of PD-(L)1 inhibitors and, subsequently, their clearance time—Schoenfeld et al. set it at 6 months. This is a strength of both recommendations, as it has been proposed that patients who develop resistance during treatment and those who relapse after completion of treatment and a treatment-free interval differ. Relapses may be termed ‘sensitive’ or ‘partially sensitive’ rather than ‘resistant’ if the treatment-free interval from therapy discontinuation to relapse is of long (or intermediate) duration [11]. Yet, in this case, it is unclear if the time limit of 3 or 6 months is optimal and remains to be validated.

### 1.2. Oligo versus Systemic Acquired Resistance

It has been proposed that the pattern of immune resistance development—reflected in the pattern of radiologic progression—may correspond to different tumor biology and affect the efficacy of immune modulation post-disease progression. Thus, a distinction can be made between oligo-acquired resistance (oligoAR) and systemic acquired resistance (sAR). OligoAR can be defined as progression in only a few lesions. Assuming that OligoAR may reflect local immune resistance with otherwise sustained anti-tumor immunity, Schoenfeld et al. performed a retrospective analysis of 1536 patients treated in the Memorial Sloan Kettering Cancer Center with PD-(L)1 blockade (without chemotherapy). They found that patients treated with ICIs who develop OligoAR in comparison to sAR have increased overall survival (OS). The authors set the limit for the distinction between oligoAR and sAR at three lesions, but whether this is the optimal cutoff for every malignancy and every treatment is not clear. When assessing the efficacy of ICI rechallenge, the pattern of resistance (OligoAR vs. sAR) to the initial IO course may be important [14].

## 2. Methods

A literature search was performed in Pubmed and Scopus using the keywords “NSCLC”, “Non-small cell lung cancer”, “Rechallenge”, “Retreatment”, “Reinitiation”, “Restart”, “Immunotherapy”, “Immune Checkpoint inhibitor”, “ICI”, “PD-1 inhibitor*”, “PD-L1 inhibitor*”, “PD”, “Progression”, “Progressive”. Additional publications were identified from cited articles and through a targeted literature search. We defined IO Rechallenge as a subsequent line of therapy with ICIs in patients who had received ICI in a previous line of treatment, either as monotherapy or as combination therapy, and experienced disease progression either during treatment or after completing the initial ICI course and a treatment-free period. Articles, including the ones on patients who discontinued the initial course of IO due to adverse events (AEs) or physician decisions, were included in this review, provided that they also included populations who had treatment progression prior to IO reintroduction. However, studies focusing only on this population of patients resuming ICIs after initial discontinuation due to AEs were not included. As this is a Narrative and not a Systematic review, it provides a non-exhaustive view of the topic.

## 3. Discussion

### 3.1. Retrospective Rw Data

So far, two Systematic Reviews and Meta-analyses by Feng et al. and Cai et al. [15,16] assessing the efficacy and safety of ICI rechallenge in patients with NSCLC have been published. An important limitation of most published retrospective studies on IO rechallenge for NSCLC is that they do not stratify patients on the basis of acquired resistance versus primary resistance, possibly leading to an underestimation of the potential efficacy of IO rechallenge had the patients been optimally selected. Furthermore, most of them do not provide explicit information on the patterns of disease progression, and they do not make the distinction between oligoAR and systemic AR. With these limitations in mind, in the following section, we present and critically review the results of 13 retrospective studies in the real world of IO rechallenge following progression either during or following completion of the initial IO course. They are presented in two separate sections, divided into those where initial IO was discontinued due to disease progression and those including mixed populations of patients, where IO was discontinued due to PD, toxicity, completion of IO course, or physician decision. The respective findings are summarized in Table 1 and Table 2.

#### 3.1.1. Cohorts of Patients Who Discontinued Initial IO Due to PD

##### Fujita et al., 2018 [17]

Fujita et al. [17] conducted a retrospective study on 12 Asian patients who received the first course of IO with nivolumab and the second course with pembrolizumab. The median PFS at rechallenge (PFS-R) was 3.1 months (range 1.2–12.6 months). The Objective Response Rate at Rechallenge (ORR-R) was 8.3%, and the Disease Control Rate at Rechallenge (DCR-R) was 41.6%. All five patients with Disease Control (PR or SD) at pembrolizumab rechallenge had PD-L1 TPS ≥ 80%. 

##### Fujita et al., 2020 [20]

This was a retrospective cohort study of 15 Asian patients who received rechallenge IO with anti-PD-1 antibodies after prior anti-PD-L1 treatment. Of the anti-PD-L1 agents used at the initial IO, 14 patients received atezolizumab, and 1 patient received durvalumab as consolidation treatment following concurrent chemoradiation. At the rechallenge, seven patients received nivolumab, and eight patients received pembrolizumab. For nivolumab, median PFS-R was 1.9 (95% CI 0.4–3.0) months; ORR-R was 0%, and DCR-R was 14.3%. For pembrolizumab, the respective numbers were 3.1 (95% CI, 1.2–12.6) months, 0%, and 37.5%. No patient had a partial or complete response at rechallenge IO. 

In this study, anti-PD-1 treatment following previous anti-PD-L1 therapy showed very poor efficacy. However, patient selection for rechallenge may have been suboptimal, considering that they had an overall poor response to prior therapy with atezolizumab (no patients had PR and only four patients had SD) and that three patients had already received a PD-1 inhibitor prior to atezolizumab, meaning that they were subsequently rechallenged with IO for the second time. 

##### Watanabe et al., 2019 [18]

This was a retrospective study of 14 patients (4.4%) conducted at seven centers in Japan. The agents administered at the first IO course were nivolumab, pembrolizumab, and atezolizumab in 11, 1, and 2 patients, respectively. At rechallenge, the respective numbers for nivolumab and pembrolizumab were nine and five. Eight patients were rechallenged with the same ICI, and six were switched. The median PFS-R was 1.6 months (95% CI: 0.8–2.6) months. ORR-R and DCR-R were 7.1% and 21.4%, respectively. The ORR-R was 12.5% among patients who received the same kind of ICIs in the first and second IO courses and 0% among patients who switched ICIs. Two of three patients who achieved disease control with rechallenge IO received radiotherapy between the first and second ICI treatments. These two patients were also the only ones who received intervening radiotherapy. In accordance with the 2020 study by Fujita et al., the rechallenge of anti-PD-1 antibody after anti-PD-L1 antibody was ineffective in two of two patients.

Paradoxically, in these cohort studies, all three patients with a BOR-R of SD had PD at the first IO course and short duration of therapy. The authors assumed that this phenomenon might be due to pseudoprogression. However, pseudoprogression in IO-treated NSCLC, in comparison to melanoma, is a relatively rare event. Although this unlikely observation cannot be precluded, it highlights a downside of most of these retrospective real-world studies, being that BOR per RECIST 1.1 is not defined by Central Review, and there is potential for interpretation error.

##### Katayama et al., 2019 [19]

This was a retrospective study of 35 patients conducted across six institutions in Japan. The agents used were nivolumab, pembrolizumab, and atezolizumab for 19 (54.3%), 12 (34.3%), and 4 (11.4%) patients at initial IO course and 5 (14.3%), 7 (20.0%) and 23 (65.7%) at rechallenge, respectively. All patients switched ICI agents at rechallenge. Median PFS-R was 2.7 (range, 1.4–3.7). ORR-R and DCR-R were 2.9% and 43%.

In the multivariate analysis, ECOG-PS ≥ 2 at rechallenge was negatively associated with PFS-R (HR 2.38, 95% CI 1.03–5.52, *p* = 0.043), and BMI > 20 was positively associated (HR 0.43, 95% CI 0.19–0.95; *p*-value = 0.036).

##### Xu et al., 2022 [21]

This was a retrospective cohort study of 40 Asian patients. The majority of patients had received the first ICI course (21 (53%)) as a combination treatment with chemotherapy. At rechallenge, 17 (43%) patients received immunotherapy combined with chemotherapy, 20 (50%) immunotherapy with chemotherapy and angiogenesis inhibitor, 10 (25%) IO with angiogenesis inhibitor, and only 3 (8%) ICI monotherapy. Median PFS-R was 6.8 months (95% CI 5.8–7.8). ORR-R was 22.5%, and DCR-R was 85.0%; BOR was PR for 9 (22.5%), SD for 25 (62.5%), and PD for 6 (15.0%) patients. In the multivariate analysis, no baseline patient or treatment characteristics were associated with PFS-R, although this study might not have been adequately powered to identify such associations.

Compared with the aforementioned studies, the majority of recruited patients in this cohort study received combined chemoimmunotherapy both at the first course IO and at the rechallenge setting, which better reflects the current clinical practice. In comparison to most of the studies mentioned so far, of rechallenge IO monotherapy, PFS-R, ORR-R, and DCR-R were numerically higher in this study, which may indicate the potential for higher efficacy with rechallenge chemoimmunotherapy.

#### 3.1.2. Cohorts of Patients Who Discontinued Initial IO Due to PD, Toxicity, or Physician Decision

##### Gettinger et al., 2018 [27]

Gettinger et al. assessed the patterns of acquired resistance in patients from nine clinical trials who developed AR to PD-1 axis inhibitor therapy. They defined acquired resistance to IO with ICI as disease progression following PR or CR by RECIST 1.1. criteria or immune-related response criteria. Out of 28 patients included in their analysis, 3 received IO rechallenge. All three received the same ICI in the first and second courses, did not receive local therapy or other systemic therapy between the two courses, were off-treatment with a durable response for more than 3 months before acquiring resistance to the first IO course, and had oligoAR. These patients more closely mirror the populations of trial post hoc analyses that will be discussed below. Two of three demonstrated durable responses to the second course IO with a PFS of 11 months and 9 months, respectively [27].

At this point, it should be noted that the writers defined all cases of disease progression after a period of response as AR, including patients who successfully completed the initial IO course with durable responses. However, as stated in the Introduction, it is debatable whether these patients should be included under the same umbrella of acquired resistance, as these relapses might be ‘partially sensitive’ rather than ‘resistant’ [11]. This possibility is endorsed by the fact that the two patients who responded to the rechallenge achieved markedly better responses to the ICI rechallenge than the average patient discussed in this section. An interesting finding of this study is the pattern of progression to the first course of IO, as all 28 patients but one experienced OligoAR (≤three progressive lesions), with Lymph nodes being the predominant site of progression [27].

##### Niki et al., 2018 [22]

This was a retrospective study of 10 Asian patients who were rechallenged with either nivolumab or pembrolizumab, following progression with prior nivolumab treatment. At rechallenge, 10 patients were retreated with nivolumab, and 1 received pembrolizumab. The median PFS-R was 2.7 (0.5–16.1) months. ORR-R was 27%, and DCR-R was 45.5%; three patients had PR, and two patients had SD. In this cohort, four out of five patients who responded to the initial IO treatment had disease control in the rechallenge setting as well. The only patient with BOR-1 of PD who achieved PR at rechallenge had received both chemotherapy and radiotherapy between the two IO courses, which led to the assumption that intervening cytotoxic therapy might be associated with the increased likelihood of rechallenge IO response. In patients who achieved disease control, the median duration from the end of initial IO treatment to rechallenge onset (ICI-free interval) was 1.6 months, whereas, for the non-responders, it was 4.7 months. Based on this finding, the authors assumed that shorter treatment-free interval between IO courses was related to better rechallenge IO efficacy, which, as we will discuss later, was contradicted in a phase-II clinical trial of nivolumab rechallenge in ICI-pretreated patients [28].

##### Kitagawa et al., 2020 [23]

This was a retrospective study of 17 Asian patients with NSCLC who were rechallenged with a different ICI following initial ICI treatment discontinuation. The reasons for discontinuing the treatment were progressive disease (PD) in 10 (58.9%) and irAEs in 7 (41.1%) patients. The median PFS-R was 4.0 (range, 0.4–8.0) months. ORR-R was 5.9%, and DCR-R was 58.8%.

##### Gobbini et al., 2020 [24]

This was a retrospective observational study conducted at 26 institutions in France of NSCLC patients rechallenged with ICI following progression after treatment discontinuation for at least 12 weeks due to disease progression (58 (40%)), toxicity (58 (40%)), or clinician decision (28 (20%)). The total number of patients in this study was 144. The agents used at initial IO treatment were anti-PD-1 for 126 (88%) patients and anti-PD-L1 for 118 (12%), while in the rechallenge setting, the corresponding proportions were 136 (94%) and 8 (6%). 

Rechallenge efficacy for the entire cohort: PFS-R was 4.4 (3–6.5) months. ORR-R was 16% and DCR-R 47%, with BOR-R being PD for 38% of patients.

Rechallenge efficacy per subgroups: When excluding patients who discontinued initial IO treatment due to toxicity, ORR-R was 13%, and DCR-R was 44%. All four efficacy measures (PFS-R, OS-R, ORR-R, DCR-R) were numerically shorter in this subgroup. 

Although PFS-R in this combined cohort was numerically higher than most of the other studies in this section, the respective PFS-R for the subgroup of 58 patients who discontinued treatment due to PD was similar [2.9 months (95% CI, 2.0–4.4)]. 

Regarding biomarkers for ICI rechallenge efficacy, only ECOG PS at rechallenge was found to be independently associated with PFS-R. However, in the univariate analyses, PFS-R was additionally positively associated with initial ICI discontinuation due to toxicity (HR = 0.54, 95% CI, 0.33–0.86); *p* = 0.02) and negatively associated with patients having received chemotherapy between the two IO courses (HR = 1.81, 95% CI, 1.21–2.72, *p* = 0.004). In this study, either, the BOR-R was not associated with BOR-1 (*p* = 1.101). This finding—in line with most of the other studies mentioned so far—supports the view that the efficacy of IO retreatment in the setting of NSCLC cannot be predicted based on the initial response per the RECIST 1.1 criteria. This is contradictory to what has been observed in melanoma patients retreated with ICIs, and it endorses the idea that radiologic criteria other than RECIST 1.1 should be explored to better predict IO rechallenge response in NSCLC [13].

##### Furuya et al., 2021 [25]

This was a retrospective study of 38 Asian patients who received atezolizumab as an IO rechallenge after previous anti-PD-1 therapy across eight institutions. The reasons for the first-course IO discontinuation were PD, toxicity, or clinician decision, although the frequency of each discontinuation reason was not reported. In contrast to most of the previously mentioned studies, the primary goal of this study was not to assess rechallenge IO efficacy; thus, information for the subpopulation of 38 patients who received atezolizumab rechallenge is relatively limited. Furthermore, the efficacy measure used was a time-to-treatment failure (TTF), defined as the time interval from ICI onset to treatment discontinuation for any cause instead of PFS. For rechallenge IO, TTF-R was 1.9 months, while ORR-R and DCR-R were 2.6% and 34.2%, with only one patient having PR.

Although it is not reported in the original paper, it is visible from the swimmer plot that a high percentage of patients (12 out of 38) who received rechallenge therapy with atezolizumab had PD as BOR at the initial ICI course. These patients most likely have primary resistance and were a priori less likely to respond to rechallenge IO. Furthermore, the median time-to-treatment failure was short, and only 10 patients had received the first course of IO for more than 6 months. These two factors may partly account for the poor efficacy of rechallenge IO reported in this cohort.

##### Ito et al., 2021 [26]

This was a retrospective multicenter study of NSCLC patients who received PD-1 inhibitors. A subgroup of 37 patients received rechallenge immunotherapy following progression during or after completion of initial ICI treatment. Ten of these patients had initially received PD-1 inhibitor treatment for >1 year without PD. Twenty-one of 37 were rechallenged with the same agent, and 16 patients switched to a PD-L1 inhibitor. 

The median PFS-R was 2.2 months (95% CI, 1.5–4.3). Interestingly, in this study, PFS-R was statistically significantly longer in patients with BOR-1 of CR or PR, (3.8 months, 95% CI, 1.5–NR vs. 1.9 months, 95% CI, 0.8–3.7, *p* = 0.04), who discontinued initial IO for reasons other than PD (6.6 months, 95% CI, 1.5–NR vs. 1.8 months, 95% CI, 1.1–2.8, *p* = 0.01), or who had PFS ≥ 3 months after the initial IO treatment discontinuation (6.6 months, 95% CI, 1.5–NR vs. 1.8 months, 95% CI, 1.4–2.8, *p* = 0.01). The PFS-R was similar for patients treated with PD-1 and PD-L1 inhibitors at rechallenge (2.3 months, 95% CI, 1.4–10.3 vs. 2.1 months, 95% CI, 1.4–4.3, *p* = 0.32).

##### Takahara et al., 2022 [29]

This was a retrospective cohort study of 24 Asian patients who received rechallenge ICI treatment. Patients had initially received either ICI monotherapy or in combination with chemotherapy and were rechallenged with ICI monotherapy. The reasons for the initial IO discontinuation were PD in 17, toxicity in 6, and physician’s decision in 1 patient. Most patients (17 of 24) switched ICI at rechallenge. No patient rechallenged with the same agent exhibited a response at rechallenge.

The ORR-R was 8.3%, and the DCR-R was 37.5%, with two patients having PR and nine having SD.

Patients with the disease control (PR or SD) had a significantly longer duration of ICI rechallenge treatment (5.04 vs. 2.54 weeks; *p* = 0.016). In line with the study by Ito et al., Takahara et al. also found an association between BOR at initial IO and rechallenge efficacy.

##### Levra et al., 2019 [30]

The final study is discussed separately, as it is the largest of the rw-studies, but it has some important differences and limitations in relation to the ones mentioned previously, which makes its results difficult to compare or draw conclusions from. Levra et al., using data from the French National Hospital discharge database, collected information on all patients receiving nivolumab treatment between 2015 and 2016. The authors considered nivolumab to be discontinued if at least three infusions were missed. For patients receiving a second course PD-1 inhibitor, they defined it as resumption if it was administered after a treatment-free interval and rechallenge if it was administered after intervening chemotherapy. In total, 1127 patients were included in the resumption group and 390 patients in the rechallenge group. Median OS after nivolumab discontinuation was 15 months (95% CI, 13.9–16.7) in the resumption group and 18.4 months (95% CI, 14.8–21.9) in the rechallenge group. Median OS was significantly longer in patients with initial nivolumab treatment duration of at least 3 months (TTF-1 > 3 months). The corresponding hazard ratios for the rechallenge group were 0.35 (95% CI, 0.22–0.56; *p* < 0.0001) for patients treated for 3–6 months and 0.19 (95% CI, 0.10–0.33; *p* < 0.000) for patients treated for ≥6 months compared to patients treated for <3 months.

Despite being by far the largest report on rechallenge ICI among NSCLC patients, it has several important limitations. First, due to the structure of the database, the information provided on baseline patient characteristics is very limited (only age, sex, and specific comorbidities). Second, the only efficacy metric provided is OS, and only for patients who died in a hospital. However, OS is not an optimal measure for the retreatment setting, as these kinds of studies inherently introduce attrition bias, considering that patients eligible for rechallenge are likely more fit and predisposed to treatment response than the broader NSCLC population. For instance, the fact that patients with TTF-1 > 3 months had improved OS might be related to the mechanism of IO resistance to initial therapy differing in these patients, or it may simply reflect the fact that these patients had a time limit set for them in which they were alive, meaning that a subset of patients with initially very poor prognosis were all included in the TTF-1 < 3 months group. Furthermore, the improved OS in patients who received intervening chemotherapy (rechallenge group) compared to patients who resumed nivolumab after a treatment-free period (resumption group) could be related to intervening chemotherapy having a positive impact on rechallenge IO efficacy, but it may as well be partly or wholly attributed to attrition bias since the former group had survived per definition for at least one more line of treatment than the latter. Third, importantly, for the purposes of our review, there is no account of the reason for the first course IO discontinuation. Thus, patients in the nivolumab resumption cohort likely represent a mixed population, with some having a treatment-free period due to adverse events but continuing therapy promptly after, without intervening PD, and some stopping due to achieving maximum benefit and then relapsing after a long treatment-free interval before retreatment with ICI. With information on the characteristics of the population under review lacking and no efficacy metrics other than OS reported, it is hard to draw conclusions about the efficacy of rechallenge immunotherapy based on this study.

#### 3.1.3. Overview

Although the cohort size of most of these studies is small, they point to the direction that rechallenge therapy with ICIs following disease progression during prior ICI treatment has overall poor efficacy in unselected patient populations. Yet, there seems to be a subpopulation of patients, not yet adequately characterized by the methods used in the published studies, who respond to IO rechallenge. Better stratification of patients in future studies based on the underlying biology of IO resistance may lead to better selection and improved rechallenge IO outcomes. Regarding the association between BOR at the first course of treatment and BOR or PFS at rechallenge, the results are contradictory. The studies by Katayama et al. [19] Gobbini et al. [24] as well as the phase-II trial by Akamatsu et al. [28] (discussed in the following section), revealed no association between BOR-1 and rechallenge IO efficacy. On the contrary, Ito et al. [26] and Takahara et al. [29] found a positive association between initial response to the IO treatment and PFS-R or longer duration of rechallenge IO, respectively. Furthermore, the best of these studies by Levra et al. [30] identified an association between TTF-1 and OS-R. In order to more safely assess the association between initial IO response and IO rechallenge efficacy, a larger-scale study with a central review of response criteria is needed. Regarding the role of intervening systemic therapy between the two courses of IO and the comparative efficacy of IO rechallenge when readministering the same agent compared to administering a different agent, the results of the cohort studies are conflicting and do not suffice to draw definite conclusions from.

### 3.2. Post Hoc Analyses of Clinical Trials

In the following section, we discuss the findings of post hoc analyses of phase-III clinical trials of NSCLC ICI-based treatment for the subpopulations of patients who were retreated with ICI following disease progression. We only include studies that report the efficacy of the results of the subsequent course of immunotherapy. The corresponding results are summarized in Table 3. It should be noted that the patient populations assessed in these studies are highly selected. They all completed the first course of immunotherapy with no progression, and then, they were rechallenged with the same ICI. As such, they represent the population that is more likely to respond to immunotherapy retreatment. Furthermore, there is no comparative arm in the post-progression setting, so it is difficult to assess whether rechallenge immunotherapy is preferable to changing the line of treatment. However, a notable number of patients, both in clinical trials and in the real world, complete the first course of immunotherapy [31]; it is worth summarizing the evidence of rechallenge efficacy in those patients.

#### 3.2.1. KEYNOTE 042

In this phase-III randomized trial, previously untreated patients with locally advanced or metastatic NSCLC and PD-L1 TPS ≥ 1% were randomized to receive either pembrolizumab monotherapy or Platinum-based chemotherapy as first-line systemic treatment. Pembrolizumab was discontinued after 35 cycles/2 years of therapy if disease progression had not occurred. Patients in the pembrolizumab arm who completed 35 cycles or stopped treatment after complete response (CR) were eligible for second-course pembrolizumab treatment.

From the pembrolizumab intention-to-treat population, 102 (16.0%) patients completed 35 cycles of treatment. Of those patients, 33 received second-course pembrolizumab. The median time from random assignment to database cutoff was 63.7 (range 52.0–75.2) months. ORR-R was 15.2%, and DCR-R was 75.8%; five patients (15.2%) had PR, and 20 (60.6%) had SD. At the data cutoff, two of them (6.1%) were alive without disease progression. PFS at subsequent course immunotherapy was not reported; however, a swimmer plot depicting the time course and response to treatment in this subpopulation can be found in the Supplementary material of the original publication [6]. The frequency of treatment-related adverse events in this subpopulation is also not reported [6].

#### 3.2.2. KEYNOTE 024

In this phase-III open-label trial, previously untreated patients with stage IV NSCLC and PD-L1 TPS > 50% were randomized to receive either pembrolizumab monotherapy or platinum-based chemotherapy as the first-line treatment. Pembrolizumab was discontinued after 35 cycles/2 years of therapy if disease progression had not occurred. Patients could receive a second course of pembrolizumab (up to 17 cycles) in case of PD, following either completion of 35 cycles of pembrolizumab or following confirmed complete responses (CR) for patients who had received at least 6 months of treatment and two more cycles of pembrolizumab after CR.

Thirty-nine of 151 patients (25.8%) in the pembrolizumab intention-to-treat population completed 35 cycles/2 years of treatment. The median (range) time from the completion of 35 cycles to the data cutoff was 34.7 months (31.2–44.1). Baseline characteristics of these patients were similar to the overall pembrolizumab intention-to-treat population, although a higher percentage had ECOG PS 0 (41.0% vs. 35.1%) and treated brain metastases (23.1% vs. 11.7%).

Twelve patients received a second pembrolizumab course after investigator-assessed PD. ORR-R was 33.3%, and DCR-R was 83.3%. All four patients (33.3%) who responded had PR, and six (50.0%) patients had SD. Treatment response per RECIST 1.1 was assessed by the investigator. At the data cutoff, eight (66.7%) patients were alive, and five patients (41.7%) had not experienced PD. Five patients (41.7%) experienced treatment-related AEs during the second course; all of them were Grade 1 or 2 [8].

#### 3.2.3. KEYNOTE 010

In KEYNOTE 010, a phase-III open-label, randomized trial, patients with stage IIIB/IV NSCLC and PD-L1 TPS > 1% were randomized to receive either pembrolizumab monotherapy or docetaxel as the second-line treatment, following progression after at least two cycles of platinum-based chemotherapy [7]. Pembrolizumab was discontinued after 35 cycles/2 years of therapy if disease progression had not occurred. In the five-year survival update, Herbst et al. reported the efficacy outcomes for patients who completed treatment with pembrolizumab and the subset of these patients who were retreated with pembrolizumab following post-therapy-completion PD. At data cutoff, 79 patients had completed 35 cycles (2 years) of pembrolizumab. Characteristics of patients who completed IO were similar to the intention-to-treat population; however, a higher percentage were <65 years old (69.6% vs. 57.2%) and had PD-L1 TPS ≥ 50% (73.4% vs. 42.0%), while a smaller percentage had received ≥ two prior lines of systemic therapy (19.0% versus 28.7%) or harbored EGFR mutations (1.3% vs. 8.8%).

At the data cutoff, 21 patients had received the second-course pembrolizumab. The ORR-R was 52.3%, and the DCR-R was 81.0%; 1 patient had CR; 10 patients had PR, and 6 patients had SD. Three patients had progressive disease at the first restaging, and eight had subsequent disease progression, of whom five had prior SD and three prior PR. Treatment response and disease progression were evaluated per RECIST 1.1 by central review. At the data cutoff, six (28.6%) patients who received second-course pembrolizumab had died. Regarding treatment-related adverse events (AEs), 10 of 21 patients (47.6%) experienced at least one rechallenge, of which two had Grade 3, including one patient with Pneumonitis. All of them had treatment-related AEs in the first IO course as well [32].

#### 3.2.4. Overview

In summary, all three KEYNOTE trials mentioned in this review (042, 024, and 010) included similar patient populations; NSCLC patients with metastatic disease or metastatic/locally advanced disease, who received rechallenge immunotherapy, were treated with pembrolizumab both at the initial randomization and upon rechallenge, had completed two years of pembrolizumab prior to retreatment, and did not receive any other systematic treatment between the two immunotherapy courses. However, there were some key differences across these three trials. Both KEYNOTE 042 and KEYNOTE 024 assessed pembrolizumab as the first-line systemic therapy for previously untreated patients, but KEYNOTE 024 only included patients with PD-L1 TPS ≥ 50%, while KEYNOTE 042 included patients with PD-L1 TPS ≥ 1%. This variation may be enough to account for the differences in ORR at a subsequent course of IO (15.2 vs. 33.3 in KEYNOTE 042 and KEYNOTE 024, accordingly). KEYNOTE 010, on the contrary, included patients treated with pembrolizumab following progression to previous line platinum-based chemotherapy. ORR to second course ICI, in this case, was numerically higher (52.3% vs. 33.3% and 15.2%) than the other two studies. A likely explanation is that patients who received rechallenge immunotherapy in this study represent a highly selected population who were intrinsically more responsive to immune modulation. This assumption is enhanced by the fact that the proportion of patients who completed two years of therapy in KEYNOTE 010 was significantly smaller than in the other two studies, particularly KEYNOTE 024, which included a similar population of PD-L1 high patients [6,8,32]. However, the possibility that prior chemotherapy affected the likelihood of response to rechallenge immunotherapy (i.e., through increased tumor neoantigen presentation and optimization of T-cell clonal differentiation during the first-course immunotherapy) cannot be disregarded.

Interestingly, although ORR-R is numerically quite different in these three trials, DCR-R is relatively similar (75.8 vs. 83.3 vs. 81.0 for KEYNOTE 042, KEYNOTE 024, and KEYNOTE 010, respectively) [6,8,32]. This is intriguing, as it may indicate that PD-L1 TPS is associated with the probability of response to immunotherapy rechallenge in this population of patients who completed treatment without progression but not associated with the probability of rapid progression. Another interesting observation is that approximately 20–25% of patients who successfully completed two years of ICI therapy exhibited rapid progression when rechallenged with the same agent. Better characterization and tissue analysis of this subpopulation with the initially durable response and subsequent resistance to IO could provide valuable insight into the mechanisms of acquired resistance to PD-1 inhibition and help more accurately identify the best candidates for the ICI rechallenge.

### 3.3. Phase-II Trial of Nivolumab Retreatment for Patients with NSCLC [28]

This open-label, multi-institutional, single-arm, phase-II trial was the first study to assess the efficacy of nivolumab rechallenge in patients with NSCLC who responded to ICI and had an ICI-free interval. Although it is a negative study, failing to meet its primary endpoint of ORR-R 20%, several interesting points should be taken into consideration. 

In this study, eligible patients needed to have had a clinical benefit at prior ICI-based treatment, defined as CR, PR, or SD, for at least 6 months and an ICI-free interval ≥ 60 days. The criterion of clinical benefit is in accordance with the SITC recommendations for defining acquired resistance to IO, although the minimum ICI-free interval was set by the investigators. As already mentioned, the primary endpoint was ORR. The sample size was calculated based on the assumption that it would provide a 10% improvement over the respective efficacy of chemotherapy in the second and third lines (20% vs. 10%). Fifty-nine patients were evaluated for nivolumab rechallenge efficacy. The majority (*N* = 54, 92%) had initially received IO monotherapy. The cause of discontinuation was irAEs in 20 patients.

ORR-R was 8.5% (95% CI, 2.8–18.7%), with five patients achieving PR. The median PFS-R was 2.6 months (95% CI, 1.6–2.8 months). Notably, however, the median PFS-R was 11.1 months for the five patients who achieved PR as BOR-R.

Interestingly, the multivariate analysis revealed that ICI-free interval was the only significant predictor of longer PFS (≤9.2 vs. >9.2 months; HR, 2.02, 95% CI, 1.10–3.73, *p* = 0.02), which might have been associated with the fact that, as stated above, patients with durable response during an adequate treatment-free interval might have had partially sensitive, instead of resistant, disease. 

A very important point to be mentioned is that the patients were heavily pretreated prior to nivolumab rechallenge, with the median number of prior chemotherapy lines being 3 (range 1–6). This translates to the median line of nivolumab rechallenge being the fourth or the fifth (taking into consideration the previous IO line as well). Although the ORR goal “was set on the basis of an assumption that retreatment with nivolumab would improve the ORR from 10% to 20% in the second- or later-line setting”, the ORR of chemotherapy (and similarly rechallenge IO) is not expected to be the same in the second and fourth/fifth line of therapy. This fact makes us cautious in the evaluation of the study results.

### 3.4. Ongoing Clinical Trials

Presently, clinical trials aiming to address treatment beyond the progression of the ICIs focus on combination therapies to overcome the acquired resistance to immunotherapy. A summary of ongoing clinical trials registered in clinicaltrials.gov is provided in Table 4. Only studies with at least one enrolled participant at the time of evaluation are included.

### 3.5. Biological Rationale—The Example of Melanoma

In order to predict response to immunotherapy rechallenge, a better understanding of the biological adaptations of the tumor and the microenvironment to immune checkpoint inhibition is required. In a study published in 2017 in *Cell*, Riaz et al. explored tumor cell and T-cell adaptations in 68 patients with melanoma treated with nivolumab; 35 previously treated with anti-CTLA-4 inhibitor Ipilimumab (Ipi-P) and 33 Ipi-naive (Ipi-N). Although cytolytic activity and response did not differ significantly between the pretreated and non-pretreated populations, T-cell dynamics in response to nivolumab treatment differed between the two groups. In Ipi-P responders, there was an increase in T-cell richness (increased number of CDR3s) without significant change in T-cell evenness, while in Ipi-N patients, there was a significant decrease in T-cell evenness (increased T-cell diversification) without significant change in T-cell richness. These results indicate that in Ipi-P patients, Tumor Infiltrating Lymphocytes (TILs) are already preselected by the tumor antigenic landscape during prior IO treatment, and resistance may arise from T-cell exhaustion through PD-1/PD-L1 signaling. This upregulation of PD-L1 expression in tumor cells and tumor-infiltrating immune cells happens largely as a response to the secretion of IFNγ and is part of the adaptive immune resistance. The increased interaction of PD-1 with PD-L1 leads to T-cell dysfunction, a phenomenon also called T-cell exhaustion [33]. So, in this population of pretreated patients, anti-PD-1 therapy works mainly by alleviating exhaustion among the existing TIL clones, while in Ipi-N patients, anti-PD-1 therapy leads to selective intratumoral expansion of tumor-reactive clonotypes. In this context, we would expect that the mechanism of resistance to the initial CTLA-4 inhibition determines the probability of response to the subsequent line immune checkpoint inhibition. If the mechanism of immune evasion is T-cell exhaustion, then IO retreatment after a time-off treatment is likely to lead to an anti-tumor response. However, if acquired resistance is driven by clonal expansion of cancer cells with subclonal mutations, patients are unlikely to respond to IO rechallenge since T-cell populations have already been selected to target neoantigens of the initial tumor population during the prior IO course. In the clinical setting, this indicates that in patients who immediately progressed during anti-CTLA-4 treatment, rechallenge is likely to be a futile strategy. Accordingly, in patients with initial partial or complete responses, IO rechallenge is likely a reasonable strategy, although more research is needed to identify biomarkers predictive of response. Finally, stable disease remains a grey zone, probably reflecting mixed cancer cell populations with differential immune responses. This diverse patient population is most in need for predictive biomarkers for IO retreatment response and might benefit more from a multimodality treatment strategy following progression, such as immunotherapy combined with chemotherapy. In the setting of melanoma, this assumption has been confirmed clinically, as a response to prior IO is related to the probability of response to rechallenge IO. However, in NSCLC, both Akamatsu et al. [28] and Gobbini et al. [24] found that the response to rechallenge IO was independent of the response to first course IO. At this point, it is important to note that the research cited above focused on patients with melanoma receiving PD-1 inhibitors who were previously treated with a CTLA-4 inhibitor, while in NSCLC, patients are rechallenged with PD-(L)1-based regimens after previous PD-axis inhibition. In this respect, the mechanism of resistance to initial IO treatment and T-cell dynamics differ in these patients, and similar research is needed in the setting of NSCLC to know if these results are applicable in this setting as well.

## 4. Conclusions

Based on current evidence, the authors would consider IO rechallenge in patients who successfully completed the first course of IO and subsequently progressed after an adequate treatment-free interval, taking into account their PS and AEs at prior IO. The data mined during this review is not sufficient to similarly educate retreatment decisions in the setting of durable response following discontinuation due to AEs. Regarding patients who progressed during the first IO treatment, the limited existing evidence does not show superior ORR with rechallenge ICIs compared to the subsequent line of chemotherapy in unselected NSCLC patient populations. Yet, as there are no properly designed prospective comparative studies of the two treatment strategies, no definite conclusions can be currently drawn in this respect. Finally, based on the current evidence, it seems that BOR per RESIST 1.1 at initial course IO is not an adequate measure of the rechallenge IO efficacy, and predictive markers are needed to guide decisions in this setting.

## 5. Future Directions

In order to determine if the underlying biology of resistance can be expressed clinically, future research on immunotherapy retreatment should include a more explicit description of radiologic progression, taking into consideration the dynamic change in each lesion rather than a gross definition of disease progression based on RECIST 1.1. criteria. For instance, oligoprogression is likely to indicate a different biological mechanism of IO resistance than systemic progression. Another interesting field of research is the role of radiotherapy in IO-pretreated patients. There are indications from small retrospective cohort studies that patients subjected to local radiotherapy respond better to IO retreatment. Comparing T-cell dynamics between patients treated with IO rechallenge who either did or did not receive prior radiotherapy would provide valuable insight into the way that radiotherapy can affect neoantigen presentation [34] and subsequent T-cell differentiation and clonal expansion. Finally, more research is needed to validate and optimize the proposed criteria for the clinical definition of acquired immunotherapy resistance in the setting of NSCLC. Some important questions to be answered in this respect are the following: 1. How do we better classify resistance in patients with SD as the best overall response to the first course of IO? 2. Should there be a minimum duration of response to distinguish between primary and acquired resistance, and if yes, what should the cutoff be? 3. Are RECIST criteria enough, or have we better accounted for tumor kinetics and individual lesion response patterns to tailor more personalized treatments in the post-IO progression setting?

## Figures and Tables

**Table 1 cancers-16-01196-t001:** Summary of ICI Rechallenge efficacy from retrospective real-world studies of patients who discontinued initial course IO due to disease progression.

Study		Fujita et al., 2018 [17]		Watanabe et al., 2019 [18]		Katayama et al., 2019 [19]		Fujita et al., 2020 [20]		Xu et al., 2022 [21]	
No of patients		12		14		35		15		40	
Discontinuation reason *		PD		PD		PD		PD		PD	
IO course		1st course	Rechallenge	1st course	Rechallenge	1st course	Rechallenge	1st course	Rechallenge	1st course	Rechallenge
Agent used		Anti-PD-1	Anti-PD-1	Anti-PD (L)-1	Anti-PD-1	Anti-PD (L)-1	Anti-PD (L)-1	Anti-PD-L1	Anti-PD-1	Anti-PD-1 ± Chemo ± anti-angio	Anti-PD (L)-1 ± Chemo ± anti-angio
Nivolumab, *N* (%)		12 (100)	12 (100)	11 (78.6)	9 (64.3)	19 (54.3)	7 (20.0)	0 (0)	8 (53.3)	NR	NR
Pembrolizumab, *N* (%)		0 (0)	0 (0)	1 (7.1)	5 (35.7)	12 (34.3)	5 (14.3)	0 (0)	7 (46.7)	NR	NR
Atezolizumab, *N* (%)		0 (0)	0 (0)	2 (14.3)	0 (0)	4 (11.4)	23 (65.7)	14 (93.3)	0 (0)	NR	NR
Durvalumab, *N* (%)		0 (0)	0 (0)	0 (0)	0 (0)	0 (0)	0 (0)	1 (6.7)	0 (0)	NR	NR
Immunotherapy-free interval		NR		NR		5.2 (3.5–7.9)		NR		NR	
Line of treatment, Median (Range)		3 (2–5)	NR	NR	NR	3 (1–15)	4 (2–19)	NR	NR	1 (1–NR)	2 (2–NR)
No of cycles, Median (Range)		12.5 (2–32)	3.5 (1–17)	NR	NR	NR	NR	5 (1–15)	Nivolumab: 4 (1–7) Pembrolizumab: (1–14)	NR	NR
PFS [Median (95% CI)], months		6.2 (2.8–13.7)	3.1 (1.2–12.6)	3.7 (1.3–7.1)	1.6 (0.8–2.6)	4 (3–4.6)	2.7 (1.4–3.7)	Atezolizumab: 2.8 Durvalumab: 6.0	Nivolumab: 1.9 (0.4–3.0) Pembrolizumab: 2.8 (0.47–13.4)	5.7 (4.1–7.2)	6.8 (5.8–7.8)
ORR, *N* (%)		7 (58.3)	1 (8.3)	3 (21.4)	1 (7.1)	12 (34.3)	1 (2.9)	0 (0)	0 (0)	14 (35)	9 (22.5)
DCR, *N* (%)		9 (75)	5 (41.6)	8 (57.1)	3 (21.4)	24 (68.6)	15 (43.0)	4 (28.6)	Nivolumab: 1/7 (14.3) Pembrolizumab: 3/8 (37.5)	33 (83)	34 (85.0)
BOR											
	CR	0 (0)	0 (0)	0 (0)	0 (0)	0 (0)	0 (0)	0 (0)	0 (0)	0 (0)	0 (0)
	PR	7 (58.3)	1 (8.3)	3 (21.4)	1 (7.1)	12 (34.3)	1 (2.9)	0 (0)	0 (0)	14 (35)	9 (22.5)
	SD	2 (16.7)	4 (33.3)	5 (35.7)	2 (14.3)	12 (34.3)	14 (40.0)	4 (28.6)	Nivolumab: 1/7 (14.3) Pembrolizumab: 3/8 (37.5)	19 (48)	25 (62.5)
	PD	3 (25)	6 (50.0)	6 (42.9)	11 (78.6)	10 (28.6)	18 (51.4)	9 (64.3)	Nivolumab: 5/7 (71.4) Pembrolizumab: 4/8 (50.0)	7 (18)	6 (15.0)

* Discontinuation reason for the initial ICI course. NR: Not reported.

**Table 2 cancers-16-01196-t002:** Summary of ICI rechallenge efficacy from retrospective real-world studies of patients who discontinued initial course IO due to disease progression, adverse events, or physician decision.

Study		Niki et al., 2018 [22]		Kitagawa et al., 2020 [23]		Gobbini et al., 2020 [24]		Furuya et al., 2021 [25]		Ito et al., 2021 [26]	
No of patients		11		17		144		38		37	
Discontinuation reason		NR		PD, Toxicity		PD, Toxicity, Physician decision		PD, Toxicity, Physician decision		Mixed	
IO course		1st course	Rechallenge	1st course	Rechallenge	1st course	Rechallenge	1st course	Rechallenge	1st course	Rechallenge
Agent used		Anti-PD-1	Anti-PD-1	Anti-PD-1	Anti-PD-L1	Anti-PD (L)-1	Anti-PD (L)-1	Anti-PD-1	Anti-PD-L1	Anti-PD-1	Anti-PD (L)-1
Nivolumab, *N* (%)		11 (100)	1 (9.1)	11 (64.7)	2 (11.8)	NR	NR	29 (76.3)	0 (0)	NR	10
Pembrolizumab, *N* (%)		0 (0)	10 (90.9)	4 (23.5)	0 (0)	NR	NR	8 (21.1)	0 (0)	NR	11
Atezolizumab, *N* (%)		0 (0)	0 (0)	2 (11.8)	15 (88.2)	NR	NR	0 (0)	38 (100)	0 (0)	16
Durvalumab, *N* (%)		0 (0)	0 (0)	0 (0)	0 (0)	NR	NR	0 (0)	0 (0)	0 (0)	0 (0)
Immunotherapy-free interval		4.2 (1.0–12.7) months.		NR		NR		NR		NR	
Line of treatment, Median (Range)		5 (3–8)	NR	2 (1–4)	3 (2–9)	2 (1–(>3))	3 (1–(>3))	NR	NR	NR	NR
PFS [Median (95% CI)], months		4.9 (0.7–18.2)	2.7 (0.5–16.1)	9.7 (0.7–34.9)	4.0 (0.4–8.0)	13 (10–16.5)	4.4 (3–6.5)	NR	NR	NR	2.2 (1.5–4.3)
ORR, *N* (%)		5 (45)	3 (27.2)	6 (35.3)	1 (5.9)	50	16	8 (21.1)	1 (2.6)	22 (59.5)	NR
DCR, *N* (%)		7 (63)	5 (45.5)	9 (52.9)	10 (58.8)	76	47	24 (63.2)	13 (34.2)	31 (83.8)	NR
BOR											
	CR	0 (0)	0 (0)	0 (0)	0 (0)	10 (7)	5 (3)	0 (0)	0 (0)	1 (0.03)	NR
	PR	5 (45)	3 (27.2)	6 (35.3)	1 (5.9)	61 (43)	18 (13)	8 (21.1)	1 (2.6)	21 (56.8)	NR
	SD	2 (18.2)	2 (18.2)	9 (52.9)	9 (52.9)	38 (26)	45 (31)	16 (42.1)	12 (31.6)	9 (24.3)	NR
	PD	4 (36.4)	6 (54.5)	2 (11.8)	7 (41.2)	26 (18)	54 (38)	11 (29.9)	19 (50)	6 (16.2)	NR

NR: Not reported.

**Table 3 cancers-16-01196-t003:** Summary of post hoc analyses of phase-III clinical trials for ICI NSCLC treatment in which a subpopulation of patients was rechallenged with the same ICI following completion of treatment and subsequent disease progression.

Trial Name (Line)		KEYNOTE 042 (First)	KEYNOTE 024 (First)	KEYNOTE 010 (Second)
Population *^1^ (selection)		1274 (PD-L1 ≥ 1%)	305 (PD-L1 ≥ 50%)	1033 (PD-L1 ≥ 1%)
Arms		(1) Pem200 mg Q3w (2) Chemo	(1) Pem200 mg Q3w, (2) Chemo	(1) Pem2 mg/kg Q2w (2) Pem10 mg/kg Q2w (3) Doce 75 mg/m2 Q3w
ORR-1, *N* (%)				
	Total population	174 (27.3) (95% CI, 23.9 to 31.0)	71 (46.1) (95% CI, 38.1 to 54.3)	Pem2 mg/kg: 62 (18) Pem10 mg/kg: 64 (18)
	PD-L1 TPS ≥ 50%	117 (39.1) (95% CI, 33.6 to 44.9)	71 (46.1) (95% CI, 38.1 to 54.3)	Pem2 mg/kg: 42 (30) Pem10 mg/kg: 44 (29)
DCR-1, *N* (%)		420 (65.9) for PD-L1 TPS ≥ 1% 206 (68.9) for PD-L1 TPS ≥ 50%.	106 (68.8)	NR
Second course ICI				
*N* out of intention-to-treat ICI patients		33 of 637	12 of 154	21 of 690
*N* out of patients who completed ICI treatment		33 of 102	12 of 39	21 of 79
Data cutoff [Median (Range)], months		63.7 (52.0–75.2) from randomization	34.7 months (31.2–44.1) from completion of first ICI course *^2^	68.1 (60.5–74.5) from randomization
ORR-R, *N* (%)		5 (15.2)	4 (33.3)	11 (52.3)
DCR-R, *N* (%)		25 (75.8)	10 (83.3)	17 (81.0)
BOR-R				
	CR, *N* (%)	0 (0.0)	0 (0.0)	1 (4.8)
	PR, *N* (%)	5 (15.2)	4 (33.3)	10 (47.6)
	SD, *N* (%)	20 (60.6)	6 (50)	6 (28.6)
	PD, *N* (%)	3 (9.1)	1 (8.3)	3 (14.3)
PD by data cutoff, *N* (%)		15 (45.	3 (25)	11 (52.3)
Death by data cutoff, *N* (%)		11 (33.3)	4 (33.3)	6 (28.6)
AEs (No of patients, %)		NR	5 (41.7)	10 (47.6)

NR: Not reported. AEs: Adverse Events. *^1^ Intention-to-treat population. *^2^ Time from randomization to IO completion was approximately 24 months.

**Table 4 cancers-16-01196-t004:** Summary of ongoing clinical trials of immunotherapy rechallenge following progression on previous ICI regimen.

NCT Number	Cancer Type	Rechallenge ICI Regimen	Phase	Primary Outcome
NCT03976375	NSCLC	Pembrolizumab + Lenvatinib	III	OS, PFS
NCT05450692	NSCLC	Durvalumab + Ceralasertib	III	OS
NCT05941897	NSCLC	Durvalumab + Ceralasertib	II	ORR
NCT03334617	NSCLC	Durvalumab + Olaparib/AZD9150/Ceralasertib/Vistusertib/Oleclumab/Trastuzumab Deruxtecan/Cediranib	II	ORR
NCT03833440	NSCLC	Durvalumab + Monalizumab/Oleclumab/Ceralasertib/ Savolitinib	II	12-week DCR
NCT05007769	NSCLC	Atezolizumab + N-803 + Ramucirumab	II	ORR
NCT03977467	NSCLC	Atezolizumab + Tiragolumab/Chemotherapy	II	ORR
NCT05781308	NSCLC	Atezolizumab + Paclitaxel + Bevacizumab	II	6-month PFS
NCT03600701	NSCLC	Atezolizumab + Cobimetinib	II	Durable Response Rate
NCT04691817	NSCLC	Atezolizumab + Tocilizumab	I/II	ORR
NCT04911166	NSCLC	Atezolizumab + Interleukin-12 Gene Therapy	I	6-month PFS
NCT04884282	NSCLC	Nivolumab + Tedopi	II	1-year OS
NCT03527108	NSCLC	Nivolumab + Ramucirumab	II	DCR
NCT04340882	NSCLC	Pembrolizumab + Docetaxel + Ramucirumab	II	6-month PFS
NCT06028633	NSCLC	Pembrolizumab + nab-Paclitaxel + Lenvatinib	II	ORR
NCT05443971	Multiple	Pembrolizumab + EDP1503	II	Safety, tolerability, ORR
NCT04725188	NSCLC	Pembrolizumab/Vibostolimab coformulation	II	PFS
NCT03881488	Multiple	Pembrolizumab + CTX-471	I	DLT, AEs, Dose
NCT05886439	NSCLC	Pembrolizumab/Durvalumab + LK101	I	DLT *, AEs
NCT05401786	NSCLC	Ipilimumab + Cemiplimab + SBRT *^2^	II	Clinical Benefit Rate
NCT06182800	NSCLC	Adebrelimab + Bevacizumab + Docetaxel	II	6-month PFS
NCT05842018	NSCLC	Toripalimab + Anlotinib + Chemotherapy	II	PFS
NCT06127303	NSCLC	Toripalimab + Cryoablation	II	PFS
NCT03228667	Multiple	PD-(L)1 inhibitor+ N-803 + PD-L1 t-haNK	II	ORR

DLT * = Dose Limiting Toxicity. SBRT *^2^ = Stereotactic Body Radiation Therapy.

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
