# Peer review of "Efficacy of NSCLC Rechallenge with Immune Checkpoint Inhibitors following Disease Progression or Relapse"

_cancers, 2024, doi:10.3390/cancers16061196_

Round 1

Reviewer 1 Report

Comments and Suggestions for Authors

Excellent review of available data on ICI rechallenge/retreatment.  The authors comprehensively and clearly review the available data on this topic. I have only very minor comments.  On page 7, there is a typo in the section on Gobinni et al., "anti-PDL" should be "anti-PD-L1"

also, the formatting of the tables could be improved - some of the lines of the tables appear to not line up correctly 

Comments on the Quality of English Language

Very good - only very mild editing likely required

Author Response

The authors would like to thank Reviewer 1 for taking the time to critically review our article and for their positive comments. We have corrected the typo. 

Reviewer 2 Report

Comments and Suggestions for Authors

The authors have compiled a review on the administration of checkpoint inhibitors in cases of treatment failure or after relapse. They do not present their own data, but analyze existing publications. In the abstract, they describe the procedure and express their hope that a better understanding of resistance to checkpoint inhibitors can be achieved in this way. The topic is relevant and is outlined in the introduction. Data analysis only yielded results when patients in the studies were differentiated according to primary or secondary resistance. In addition, the classification according to systemic or oligo-resistance was helpful. Methodologically, this is a pure meta-analysis. In the end, only 13 usable studies could be considered, which are presented individually below. Ultimately, it appears that a rechallenge with checkpoint inhibitors only makes sense if an initial therapy has been successfully completed. 

Author Response

The authors would like to thank Reviewer 2 for their insightful feedback. 

Regarding the classification of our review, we want to clarify that while the methodology resembles aspects of a systematic review meta-analysis, it aligns more closely with a narrative review. We did not perform a pooled analysis of the results of the individual studies, as typically done in a meta-analysis, nor did we strictly adhere to inclusion/exclusion criteria characteristic of a systematic review, rather, after an extensive literature search, we narrowed down the relative studies and critically discussed each one separately.

Reviewer 3 Report

Comments and Suggestions for Authors

The authors of the manuscript entitled, “Efficacy Of NSCLC Rechallenge with Immune Checkpoint Inhibitors Following Disease Progression or Relapse” reviewed current evidence related to immunotherapy rechallenge upon disease progression or relapse. The lack of clear definition of extent of disease to distinguish oligoAR or systemicAR and timeline to define retreatment or rechallenge might have been challenging factors for the authors performing literature review. However, authors did excellent job to provide overview of retrospective data and follow up studies from large phase III studies for patients with NSCLC. I have few comments for the authors to consider.

1.     Type editing: Page 5, Fujita et al, third row, there is two “at At” after “used”.

Page 6, 3.1.2 Gettinger et al, reference #22 (double space after 2018).

Page 9, 2nd paragraph, after the “However” there is two commas.

2.     I hope that authors may able to discuss the strategies related to immunotherapy rechallenge with double immunotherapy or other combinatorial approach. (see comment #4 as well regarding ongoing clinical studies)

3.     Immunotherapy rechallenge after another line of treatment for those who had first line immunotherapy failure could be an option to rechallenge patients with immunotherapy. I hope that authors may be able to comment on this approach.

4.     Currently, there are numerous clinical trials are underway to test immunotherapy rechallenge by combining with other immunotherapy agents or targeted agents. Hope that authors may be able to provide brief summary of these trials which we hope to learn much about tumor microenvironment reprogramming to re-sensitize patients to immunotherapy.  

Author Response

The authors would like to deeply thank Reviewer 3 for their time to critically review our manuscript and for their valuable comments and suggestions.

Answer to Comment 1:

 Answer to omments 2,4: Thank you very much for this useful advice. As instructed, we have added a section on ongoing clinical trials of immunotherapy rechallenge in NSCLC (Section 3.4).

Answer to Comment 3: Thank you for this important observation: Some of the trials discussed, immunotherapy rechallenge was not strictly administered as the immediate subsequent line post initial immunotherapy failure, but rather after one or more intervening lines of chemotherapy. However, the authors do not make the distinction between patients who received intervening chemotherapy and those who received immunotherapy rechallenge immediately after.